# HiPO-MILP: Hierarchical Preference Optimization for MILP Solving

## Abstract

Mixed-integer linear programming (MILP) is a fundamental yet computationally challenging optimization problem in operations research. To accelerate the solving process, recent machine learning methods predict an initial solution and confine the subsequent search to a local trust region. However, these models face two critical challenges during training. First, the models are typically trained on a collection of high-quality solutions weighted by their objective values, which fails to account for a solution's distant to the near-optimal region and leads to a biased training signal. Second, weighting by objective value provides an ambiguous preference signal, which prevents the model from learning to explicitly distinguish between high-quality and local optimal solutions. To address the challenges, we introduce HiPO-MILP, a novel Hierarchical Preference Optimization framework. Our key idea is to define a quality score for each solution that combines its objective value with its distance to the convex hull of optimal solutions. Based on this score, HiPO-MILP constructs a three-tiered preference hierarchy that distinguishes between near-optimal, high-quality, and perturbed solutions, thereby providing a clear and robust learning signal. By training with explicit preference pairs derived from this hierarchy, HiPO-MILP learns to navigate the solution space towards regions that are not only high-scoring but also structurally closer to the global optimum. Experiments demonstrate that HiPO-MILP substantially improves solving efficiency across a diverse range of MILP benchmarks.

## 1 Introduction

Mixed-Integer Linear Programming (MILP) is a foundational model for combinatorial optimization, with widespread applications in fields such as engineering (Husseinzadeh Kashan & Ozturk, 2022), daily scheduling (Liu et al., 2021), and operations research (Golmohamadi, 2022). Traditional MILP solvers, which typically rely on the branch-and-bound algorithm, must navigate a vast and discrete solution space to find the optimal solution. This process presents a significant computational challenge, as many MILP problems are NP-hard, with search spaces that expand exponentially with the problem size. In many real-world scenarios, however, practitioners frequently encounter specific families of MILP instances that share underlying structural and statistical patterns. This has created a fertile ground for machine learning (ML) techniques designed to capture these data-driven patterns and accelerate the solving process (Gasse et al., 2019; Bengio et al., 2021). Among the various ML-enhanced solving paradigms, direct solution prediction has gained significant popularity in recent years. These approaches, including neural diving (Nair et al., 2020) and predict-and-search (PS) (Han et al., 2023), utilize a neural network to generate an initial solution for a given MILP instance. Subsequently, a traditional solver is used to find the optimal solution, but its search is confined to a predefined trust region around this initial prediction. By dramatically narrowing the search space, this method holds the promise of substantial speedups.

However, the PS paradigm depends critically on the quality of the training data—and more importantly, on the quality score that defines "quality" for such data. Due to the NP-hard nature of MILP, training labels are typically not true optimal solutions but a collection of high-quality, near-optimal solutions obtained by solvers within a time limit (Han et al., 2023; Huang et al., 2024). Existing PS methods typically select the top solutions and assign weights for training based on their objective values. We identify two key challenges in the solving process. (1) Using a solution's objective value as the sole criterion is inadequate, as it does not account for the solution's structural proximity

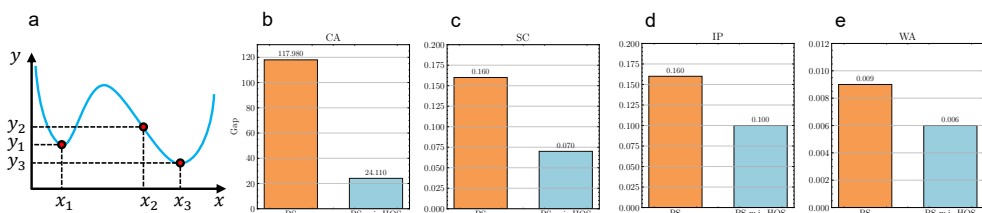

Figure 1: (a) We observe that while some solutions exhibit high-quality objective values (e.g., $x_1$), their distance from the optimal solution $x_3$ is significant. Utilizing these solutions for training may introduce additional noise. (b)-(e) present a comparison of various weighting strategies, specifically using either the objective value or the holistic quality score, to train the PS model across four datasets. Training with the holistic quality score results in a smaller performance gap. H.Q.S denotes the holistic quality score. The detailed results are reported in Appendix E.

to the true optimal region. Our analysis indicates that many solutions, even with similar objective values, can be geometrically distant from one another (see Figure 1). (2) Second, simply weighting solutions by their objective values provides an ambiguous learning signal. It fails to teach the model to explicitly distinguish between structurally superior solutions and those that are merely locally optimal. Consequently, training a model with these structurally misleading labels provides a noisy and biased learning signal, which reduces sample efficiency and often traps the solver in a local optimum, fundamentally limiting the quality of the final solution.

To address this challenge, we propose a novel metric called the holistic quality score, which evaluates solutions based on both their objective values and their internal structures. We introduce the concept of the near-optimal convex hull, defined as the convex hull of all solutions with the highest objective values. Using this convex hull, the holistic quality score for any solution is calculated as a weighted sum of its objective value and its distance to the near-optimal convex hull. This score enables us to more accurately assess the effectiveness of any given solution in the training process.

Building on the proposed holistic quality score, we introduce HiPO-MILP, a novel Hierarchical Preference Optimization approach for training solution prediction models. HiPO-MILP constructs a multi-tiered preference system based on solution quality scores. The highest preference is assigned to solutions that lie within the near-optimal convex hull, which achieve the top holistic quality scores and objective values in the dataset. Medium preference is given to solutions with high holistic quality scores that fall outside the near-optimal convex hull. The third level of preference is designated for perturbed solutions, which are further classified into two types: low-quality feasible perturbed solutions, differentiated by their holistic quality scores, and infeasible perturbed solutions, which receive the worst possible holistic quality score. For modeling training, we leverage the direct preference learning (DPO) (Rafailov et al., 2023) on explicit preference pairs derived from this clearly defined hierarchy. The preference learning framework enables our model to identify the subtle structural features that set optimal or near-optimal solutions apart from those that are merely locally optimal. Experimental results demonstrate that our method achieves state-of-the-art performance across various challenging MILP benchmarks.

We summarize the contribution of this paper as follows. (1) We propose a novel hierarchical preference optimization framework for MILP, grounded in a quality score that unifies a solution's objective value and its distance to the optimal set. (2) We integrate this preference learning strategy with a powerful prediction model to generate superior warm-start solutions that lead to faster convergence for MILP solvers. (3) We conduct extensive experiments across a diverse range of MILP benchmarks, demonstrating that HiPO-MILP substantially improves both final solution quality and solving efficiency over existing baselines.

## 2 RELATED WORKS

### 2.1 MACHINE LEARNING FOR MILPS

Researchers have leveraged machine learning to accelerate the solving process of MILP (Bengio et al., 2021). Existing learning-based solving methods include two main lines of research. The first

line uses learning-based methods to improve modules in traditional MILP solvers (Li et al., 2024). They usually replace the heuristics in the solvers with learning-based networks, such as branching (Gasse et al., 2019; Khalil et al., 2022; Kuang et al., 2024), node selection (He et al., 2014; Liu et al., 2024), cutting in branch-and-bound solvers (Wang et al., 2023; Puigdemont et al., 2024), and large neighborhood search solvers (Huang et al., 2023; Sonnerat et al., 2021).

The second line of research uses neural networks to directly predict solutions for the solvers. The solution prediction methods have gained significant popularity in recent years (Nair et al., 2020; Han et al., 2023). These methods first leverage neural networks to predict an initial solution and then call a traditional solver to search for the optimal solution around the initial solution. This line of research includes neural diving (Nair et al., 2020), predict-and-search (Nair et al., 2020), and their variants. Neural diving directly fixes some variables in a predicted partial solution, while predict-and-search employs a trust region search to allow for better feasibility. Contrastive predict-and-search (Huang et al., 2024) utilizes contrastive learning to improve the prediction accuracy. SymILO (Chen et al., 2024) exploits the symmetric features of the MILP instances to enhance the prediction capability further. DiffILO (Geng et al., 2025) integrates gradient information into the prediction model. Apollo-MILP (Liu et al., 2025), from a search perspective, leverages a prediction-correction framework to achieve better solution quality.

## 2.2 PREFERENCE OPTIMIZATION

Direct Preference Optimization (DPO) (Rafailov et al., 2023) is a stable and efficient policy-learning algorithm designed to align models with human or objective preferences, emerging as a popular alternative to Reinforcement Learning from Human Feedback (RLHF). Instead of training a separate reward model to guide the policy, DPO directly optimizes the policy using a preference dataset (Meng et al., 2024; Ethayarajh et al., 2024). Preference optimization has been applied to combinatorial optimization for enhanced exploration (Pan et al., 2025; Liao et al., 2025; Fan et al., 2025). However, these methods are tailored to routing problems, while we are focusing on the complex general MILP problems.

## 3 PRELIMINARIES

### 3.1 MIXED-INTEGER LINEAR PROGRAMMING

Mixed-integer linear programming can formulate a large family of operations research and combinatorial optimization problems, which takes the form of

$$\min_{\boldsymbol{x} \in \mathbb{Z}^p \times \mathbb{R}^{n-p}} \boldsymbol{c}^\top \boldsymbol{x}, \ \text{s.t. } \boldsymbol{A}\boldsymbol{x} \leq \boldsymbol{b}, \ \boldsymbol{l} \leq \boldsymbol{x} \leq \boldsymbol{u}, \tag{1}$$

where $\boldsymbol{x}$ is the decision variables, $\boldsymbol{c}$ is the objective coefficients, $\boldsymbol{A}$ is the constraint coefficient matrix, $\boldsymbol{b}$ is the right-hand-side terms of the constraints, and $\boldsymbol{l}$ and $\boldsymbol{u}$ are the lower and upper bound of the decision variables, respectively. To encode a MILP instance, existing learning-based methods often present the MILP instances as bipartite graphs. Each group of nodes represents the constraints and variables in the instances, respectively. Then, these methods employ graph neural network (GNN) for graph representation learning.

### 3.2 DIRECT PREFERENCE OPTIMIZATION

DPO (Rafailov et al., 2023) constructs a preference dataset for preference learning. This dataset $D$ consists of triplets $(\mathcal{I}, \boldsymbol{y}^w, \boldsymbol{y}^l)$, where for a given input $\mathcal{I}$, $\boldsymbol{y}^w$ is the preferred (winner) output and $\boldsymbol{y}^l$ is the rejected (loser) output. The core assumption of DPO is that the preference labels are generated by an implicit reward model $r^*(\boldsymbol{y}, \mathcal{I})$, where a human prefers $\boldsymbol{y}^w$ over $\boldsymbol{y}^l$ if $r^*(\boldsymbol{y}^w, \mathcal{I}) > r^*(\boldsymbol{y}^l, \mathcal{I})$.

Simple preference optimization (SimPO) (Meng et al., 2024) is an improved version of DPO with much popularity. The objective of SimPO is to train a policy $\pi_\theta$ that best satisfies these preferences. The SimPO loss function is derived from the Bradley-Terry model (Hunter, 2004), which links preferences to the underlying reward, and is expressed as:

$$L_{\text{SimPO}} = -\mathbb{E}_{(\mathcal{I}, \boldsymbol{y}^w, \boldsymbol{y}^l) \sim D} \left[ \log \sigma \left( \frac{\beta}{|\boldsymbol{y}^w|} \log \pi_\theta(\boldsymbol{y}^w | \mathcal{I}) - \frac{\beta}{|\boldsymbol{y}^l|} \log \pi_\theta(\boldsymbol{y}^l | \mathcal{I}) \right) \right]. \tag{2}$$

# 4 HiPO-MILP: HIERARCHICAL PREFERENCE OPTIMIZATION FOR MILPS

In this part, we present our HiPO-MILP framework. The overview of the framework is in Figure 2.

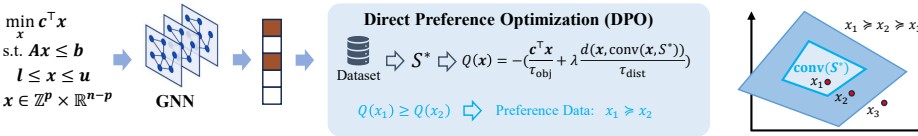

| (a) Overview of Our Framework | (b) Preference Levels |

Figure 2: (a) Overview of Our Framework. We employ a GNN to predict an initial solution, thereby accelerating the solving process for MILP. Using a training dataset that includes multiple solutions, we construct the near-optimal convex hull and calculate the holistic quality score for each solution. This score enables us to create preference pairs for preference optimization. (b) Three-Tiered Preference Levels. The preference levels are categorized from high to low: (1) solutions within the near-optimal convex hull, (2) solutions outside the convex hull with high holistic quality scores, and (3) solutions with low scores.

## 4.1 PREDICT-AND-SEARCH FOR MILPS

To approximate the solution distribution of a given MILP instance, we adopt the PS paradigm (Han et al., 2023). Specifically, this distribution is defined via an energy function, which assigns lower energy values to high-quality feasible solutions and infinite energy to infeasible ones. Mathematically, the distribution and energy function are formulated as follows,

$$p(\boldsymbol{x} \mid \mathcal{I}) = \frac{\exp\left(-E(\boldsymbol{x} \mid \mathcal{I})\right)}{\sum_{\boldsymbol{x}'} \exp\left(-E(\boldsymbol{x}' \mid \mathcal{I})\right)}, \quad \text{where } E(\boldsymbol{x} \mid \mathcal{I}) = \begin{cases} \boldsymbol{c}^\top \boldsymbol{x}, & \text{if } \boldsymbol{x} \text{ is feasible}, \\ +\infty, & \text{otherwise}. \end{cases} \quad (3)$$

Here, $\boldsymbol{x}$ denotes a candidate solution, $\mathcal{I}$ represents the MILP instance, $\boldsymbol{x}'$ are summed over the training dataset, and $\boldsymbol{c}^\top \boldsymbol{x}$ is the objective function of the MILP, ensuring that feasible solutions with better (lower) objective values correspond to lower energy. Our objective is to learn a parameterized distribution $p_{\boldsymbol{\theta}}(\boldsymbol{x} \mid \mathcal{I})$ (with parameters $\boldsymbol{\theta}$) that approximates the true solution distribution of $\mathcal{I}$. To make the learning process computationally tractable (given the discrete and high-dimensional nature of MILP solutions), we introduce a key simplification: we assume a fully factorized solution distribution over the binary variables of the MILP. This means the joint probability of a solution can be decomposed into the product of marginal probabilities for individual binary variables $p_{\boldsymbol{\theta}}(\boldsymbol{x} \mid \mathcal{I}) = \prod_{i=1}^{p} p_{\boldsymbol{\theta}}(\boldsymbol{x}_i \mid \mathcal{I})$, where $p$ is the number of binary variables in $\mathcal{I}$, and $p_{\boldsymbol{\theta}}(\boldsymbol{x}_i \mid \mathcal{I})$ denotes the model's predicted marginal probability that the $i$-th binary variable $\boldsymbol{x}_i$ takes the value 1.

To compute these marginal probabilities, we employ a Graph Neural Network (GNN) as the predictive model. The GNN takes the MILP instance $\mathcal{I}$ (typically encoded as a bipartite graph of variables and constraints) as input and outputs a $p$-dimensional vector $\hat{\boldsymbol{x}} = f_{\boldsymbol{\theta}}(\mathcal{I}) = (\hat{\boldsymbol{x}}_1, \cdots, \hat{\boldsymbol{x}}_p)^\top \in [0, 1]^p$. Each element $\hat{\boldsymbol{x}}_j$ in this vector directly corresponds to the predicted marginal probability $p_{\boldsymbol{\theta}}(\boldsymbol{x}_j = 1 \mid \mathcal{I})$ for the $j$-th binary variable. For training the GNN, we use a set of weighted feasible solutions $\{\boldsymbol{x}^{(i)}\}_{i=1}^{N}$ as supervision signals. Each solution $\boldsymbol{x}^{(i)}$ is assigned a weight proportional to the exponential of its negative objective value: $w_i \propto \exp\left(-\boldsymbol{c}^\top \boldsymbol{x}^{(i)}\right)$. This weighting scheme ensures that solutions with better objective values exert a larger influence on the training process. The training loss is a Binary Cross-Entropy (BCE) loss, which minimizes the discrepancy between the GNN's predicted marginal probabilities $\hat{\boldsymbol{x}}_j$ and the ground-truth binary values $x_j^{(i)}$ of the supervised solutions:

$$\mathcal{L}_{\text{BCE}}(\boldsymbol{\theta} \mid \mathcal{I}) = -\sum_{i=1}^{N} \sum_{j=1}^{p} w_i \cdot \left[ \boldsymbol{x}_j^{(i)} \log \hat{\boldsymbol{x}}_j + (1 - \boldsymbol{x}_j^{(i)}) \log(1 - \hat{\boldsymbol{x}}_j) \right]. \quad (4)$$

At inference time, the trained GNN outputs the vector of predicted marginal probabilities $\hat{\boldsymbol{x}} \in [0, 1]^p$. To find the optimal feasible solution for $\mathcal{I}$, we use a standard MILP solver (e.g., Gurobi or SCIP)

and constrain its search to a local trust region around $\hat{x}$—this avoids the solver exploring the entire exponentially large solution space. The trust region problem solved by the solver is defined as:

$$\min_{\boldsymbol{x} \in \mathbb{Z}^p \times \mathbb{R}^{n-p}} \left\{ \boldsymbol{c}^\top \boldsymbol{x} \mid \boldsymbol{A}\boldsymbol{x} \le \boldsymbol{b}, \, \boldsymbol{l} \le \boldsymbol{x} \le \boldsymbol{u}, \, \boldsymbol{x}_{1:p} \in \mathcal{B}(\hat{\boldsymbol{x}}, \Delta) \right\}, \tag{5}$$

where $\boldsymbol{x}_{1:p}$ denotes the subset of binary variables in $\boldsymbol{x}$, $n$ is the total number of decision variables (binary and continuous), and $\boldsymbol{A}, \boldsymbol{b}, \boldsymbol{l}, \boldsymbol{u}$ are the constraint matrix, constraint right-hand-side terms, and variable bounds of the MILP, respectively. The trust region $\mathcal{B}(\hat{\boldsymbol{x}}, \Delta)$ is defined using the $L_1$-norm: $\mathcal{B}(\hat{\boldsymbol{x}}, \Delta) = \{\boldsymbol{x} \in \mathbb{R}^n \mid \|\boldsymbol{x}_{1:p} - \hat{\boldsymbol{x}}\|_1 \le \Delta\}$, ensuring the solver only explores solutions whose binary variables are close to the GNN's predictions .

## 4.2 HOLISTIC QUALITY SCORE

Existing PS methods typically select the top solutions and assign training weights based solely on their objective values. However, relying solely on a solution's objective value is insufficient, as it does not account for the structural proximity of the solution to the true optimal region. Our analysis reveals that many solutions, despite having similar objective values, can be geometrically distant from one another. To address this, we introduce the concept of the holistic quality score, which evaluates the usefulness of each solution in the training process.

We begin by presenting a theorem that provides a geometric perspective on this issue. Let $S^*$ denote the set of all solutions with the best objective values in a given training dataset, and define their convex hull as the near-optimal convex hull, denoted by $\mathrm{conv}(S^*)$. The near-optimal convex hull has the following critical property (please see Appendix A.1 for the proof).

**Theorem 4.1.** *For a MILP problem, let $S^*$ be the set of all its optimal solutions or solutions with the best objective value in a given solution set. Let $z^*$ be the corresponding objective value. Any feasible solution $\boldsymbol{y}$ within the convex hull of $S^*$, denoted $\mathrm{conv}(S^*)$, has the objective value $z^*$.*

The distance between any solution and the near-optimal convex hull is well-defined, as stated in the following theorem (please see Appendix A.2 for the proof),

**Theorem 4.2.** *For any feasible solution $\boldsymbol{y}$ (or any point $\boldsymbol{y}$), the distance from $\boldsymbol{y}$ to the set $\mathrm{conv}(S^*)$ is uniquely determined. That is, there exists a unique distance value $d(\boldsymbol{y}, \mathrm{conv}(S^*))$.*

The foundation of our preference framework is a holistic quality score, $Q(\boldsymbol{x})$, designed to more accurately measure a solution's true value as a training label. Instead of relying solely on the objective, this score combines two critical components (1) objective value and (2) structural proximity, which is the distance of a solution $\boldsymbol{x}$ to the near-optimal convex hull. This metric captures how structurally similar a solution is to the optimal or near-optimal region. The score is defined as follows: for any solution $\boldsymbol{x}$, we have

$$Q(\boldsymbol{x}) = \begin{cases} -\left( \dfrac{\boldsymbol{c}^\top \boldsymbol{x}}{\tau_{\mathrm{obj}}} + \lambda \dfrac{d(\boldsymbol{x}, \mathrm{conv}(S^*))}{\tau_{\mathrm{dist}}} \right), & \text{if } \boldsymbol{x} \text{ is feasible;} \\ -\infty, & \text{otherwise,} \end{cases} \tag{6}$$

where $\lambda$ is a non-negative constant that balances the contributions of the objective and the distance terms, and $\tau_{\mathrm{obj}}, \tau_{\mathrm{dist}}$ are normalizing factors. In particular, $\tau_{\mathrm{obj}}$ serves two purposes: it scales the objective value to be comparable with the distance term, and it also unifies the optimization direction. Specifically, for maximization instances, $\tau_{\mathrm{obj}}$ is set to a negative value so that the objective term is effectively minimized, thereby aligning it with the minimization perspective of the distance term; for minimization instances, $\tau_{\mathrm{obj}}$ is positive and the direction is preserved. The following observation shows that the solution of the near-optimal set has the highest holistic quality score (please see Appendix A.3 for the proof).

**Theorem 4.3.** *Given a solution set $S$, the solutions in the corresponding near-optimal set $S^*$ has the lowest holistic quality score, i.e., for any $\boldsymbol{x}, \boldsymbol{y} \in S$ and $\boldsymbol{x} \in S^*$, we have $Q(\boldsymbol{x}) \ge Q(\boldsymbol{y})$.*

## 4.3 HIERARCHICAL PREFERENCE OPTIMIZATION

Using the quality score $Q(\boldsymbol{x})$, we categorize the pool of solutions collected from the solver into a three-tiered hierarchy, transforming a flat set of solutions into structured data that is well-suited for preference learning. We introduce the following hierarchical preference framework:

**Level 1: Solutions in the Near-Optimal Convex Hull (High Preference)**. At the top of the hierarchy are the solutions with the highest holistic quality scores. We assign them the highest preference to encourage the model to closely approximate their structure. Accurate predictions in this optimal region significantly narrow the search space for solvers. We denote this set of solutions as $S^*$.

**Level 2: High-Quality Feasible Solutions (Medium Preference)**. The second tier comprises solutions with high holistic quality scores that lie outside the near-optimal convex hull. We refer to this set of solutions as $S_2$.

**Level 3: Perturbed Solutions (Low Preference)**. The final tier consists of perturbed solutions, which serve as explicit negative examples. These solutions exhibit low holistic quality scores and include both low-quality feasible perturbations and infeasible solutions, with the latter receiving the lowest preference. We denote this set of solutions as $S_3$.

By constructing preference pairs from these tiers (for instance, a Level 1 solution is preferred over a Level 2 solution, and a Level 2 solution is preferred over a Level 3 solution), we create a rich training dataset. This framework enables HiPO-MILP to learn a more nuanced and effective policy for predicting high-quality, warm-start solutions.

## 4.4 PREFERENCE LEARNING IN MILP

HiPO-MILP utilizes simple preference optimization (SimPO) for hierarchical preference learning. First, we define an implicit reward $r_\theta(\boldsymbol{x})$ that the model assigns to a given solution $\boldsymbol{x}$. This reward is calculated as the log-probability of the solution under the model's current policy, scaled by a temperature parameter $\beta$:

$$r_\theta(\boldsymbol{x}) = \beta \log \pi_\theta(\boldsymbol{x}|\mathcal{I}) = \beta \sum_{j=1}^{p} \log \pi_\theta(\boldsymbol{x}_j|\mathcal{I}). \tag{7}$$

A higher reward indicates that the solution is more likely according to the model's learned distribution. The goal of preference learning is to adjust the model's parameters $\theta$ such that the rewards align with our hierarchical preference structure.

To construct preference pairs for training, we sample a "positive set" of preferred solutions and a "negative set" of rejected solutions, resulting in a set of preference pairs $\mathcal{D} = \{(\boldsymbol{x}^{w,(k)}, \boldsymbol{x}^{l,(k)})\}_k$. This sampling reflects our three-tiered hierarchy: if $\boldsymbol{x}^{w,(k)} \in S^*$, then $\boldsymbol{x}^{l,(k)} \in S_2 \cup S_3$; if $\boldsymbol{x}^{w,(k)} \in S_2$, then $\boldsymbol{x}^{l,(k)} \in S_3$. The HiPO loss is formulated as follows:

$$\mathcal{L}_{\text{HiPO}} = -\mathbb{E}_{(\mathcal{I}, \boldsymbol{x}^w, \boldsymbol{x}^l) \sim D} \left[ \log \sigma \left( \frac{\beta}{|\boldsymbol{x}^w|} \log \sum_{i=1}^{|\boldsymbol{x}^w|} \pi_\theta(\boldsymbol{x}_i^w|\mathcal{I}) - \frac{\beta}{|\boldsymbol{x}^l|} \log \sum_{i=1}^{|\boldsymbol{x}^l|} \pi_\theta(\boldsymbol{x}_i^l|\mathcal{I}) - \gamma \right) \right]. \tag{8}$$

Here, $\gamma$ represents trainable margins that help create separation between the reward distributions of different solution tiers.

The overall training objective for HiPO-MILP combines the standard Binary Cross-Entropy (BCE) imitation learning loss with our new preference loss:

$$\mathcal{L}_{\text{total}} = \mathcal{L}_{\text{BCE}} + \eta \mathcal{L}_{\text{HiPO}}, \tag{9}$$

where $\eta$ is a hyperparameter that balances the two terms. The term $\mathcal{L}_{\text{BCE}}$ acts as a stabilizer, ensuring that the model learns the general distribution of high-quality solutions. In contrast, $\mathcal{L}_{\text{HiPO}}$ refines this distribution by imposing the crucial hierarchical preference structure. This dual-objective approach results in a more robust and accurate prediction model capable of generating superior warm-start solutions.

## 5 EXPERIMENTS

In this section, we conduct extensive experiments to validate the effectiveness of HiPO-MILP. Specifically, these experiments include an analysis of HiPO-MILP's performance improvements over traditional solvers (Section 5.2), alongside an evaluation of its generalization ability (Appendix F.3), and detailed ablation studies (Section 5.3 and Appendix F.2) to highlight the efficiency of key design components in our framework.

### 5.1 EXPERIMENT SETTINGS

**Benchmarks**  Our experimental evaluations base on four popular MILP benchmarks within the ML4CO field: combinatorial auctions (CA) (Leyton-Brown et al., 2000), set covering (SC) (Balas & Ho, 1980), item placement (IP) (Gasse et al., 2022), and workload appointment (WA) (Gasse et al., 2022). The first two, CA and SC, introduced in (Gasse et al., 2019), serve as standard testbeds commonly adopted for evaluating ML solver performance (Gasse et al., 2019; Han et al., 2023; Huang et al., 2024; Liu et al., 2025), while the latter two, IP and WA, derive from two challenging real-world problem families featured in the NeurIPS ML4CO 2021 competition (Gasse et al., 2022). Our experimental setup includes 240 training instances, 60 validation instances, and 100 testing instances, which aligns with the configurations outlined in Han et al. (2023). We give a more detailed information about benchmarks in Appendix C.

**Baselines**  We primarily compare our method against two categories of baseline methods. First, we include traditional solvers, including Gurobi (Gurobi Optimization, 2021) and SCIP (Achterberg, 2009), to evaluate whether the heuristic solutions generated by HiPO-MILP can enhance solving performance. Second, we compare our method with two classic ML-based methods: Predict-and-Search (PS) (Han et al., 2023) and Contrastive Predict-and-Search (ConPS) (Huang et al., 2024). PS is a milestone work, as it was the first to propose predicting a partial solution via ML and then performing a search to obtain a high-quality solution. ConPS, by contrast, is a stronger baseline that uses contrastive learning between high-quality and low-quality solutions. Detailed implementation details for these baselines are provided in Appendix B.

**Metrics**  We evaluate each method on all test instances and report the best objective value OBJ attained within a 1,000-second time limit. Consistent with the experimental setup in Han et al. (2023), we run a single-threaded instance of Gurobi with a 3,600-second time limit, and designate its resulting best objective value as the best-known solution (BKS). The BKS is used to approximate the true optimal value of the problem. We define the absolute primal gap as the absolute difference between the best objective value identified by each solver and the BKS, formally expressed as $\text{gap}_{\text{abs}} := |\text{OBJ} - \text{BKS}|$. For identical solving time limits, a smaller absolute primal gap indicates superior performance.

**Implementations**  In our experiments, we first construct a near-optimal convex hull from the highest-quality solutions during dataset collection. To generate perturbed solutions, we follow the method proposed in ConPS (Huang et al., 2024): specifically, we randomly perturb a small fraction of binary variables in these high-quality solutions by flipping their values. These perturbed solutions are mostly infeasible or of low quality. We then compute the holistic quality score for each solution based on Equation 6. We provide a detailed description of the implementation, including data generation, training, and inference. Please refer to Appendix B for more details.

### 5.2 MAIN EVALUATION

**Solving Performance**  To evaluate the effectiveness of the proposed HiPO-MILP framework, we compare its solution performance against baseline methods under a 1,000-second time limit. Table 1 presents two core metrics for each solver: the average best objective value and the average absolute primal gap. Given the use of challenging large-scale benchmark instances, all solvers exhausted the 1,000-second time limit during experiments, this ensures a fair comparison of performance within the same computational budget. The CA, SC, IP, and WA datasets comprise problems of varying complexity, with IP and WA instances in particular exhibiting more intricate structures, thereby posing greater challenges to solver performance. Among the baseline methods, ConPS consistently out-

Table 1: Performance comparison of between HiPO-MILP and baseline methods under a 1000-second time limit. All ML-based methods are built upon Gurobi and SCIP solvers, respectively. Given the use of challenging large-scale benchmark instances, every solver involved hits the predefined 1000s time limit during experiments. We therefore report two core metrics: the average best objective value and the primal gap ($gap_{abs}$). '↑' denotes that higher is better, while '↓' indicates that lower is better. We mark the **best values** in bold. Furthermore, the improvement of HiPO-MILP over traditional solvers is quantified, with the improvement degree evaluated based on $gap_{abs}$.

| | CA (BKS 97375.08) | | SC (BKS 124.95) | | IP (BKS 11.21) | | WA (BKS 703.09) | |
|---|---|---|---|---|---|---|---|---|
| | Obj ↑ | $gap_{abs}$ ↓ | Obj ↓ | $gap_{abs}$ ↓ | Obj ↓ | $gap_{abs}$ ↓ | Obj ↓ | $gap_{abs}$ ↓ |
| Gurobi | 97247.84 | 127.24 | 125.42 | 0.47 | 11.32 | 0.11 | 703.27 | 0.18 |
| PS+Gurobi | 97257.10 | 117.98 | 125.11 | 0.16 | 11.37 | 0.16 | 703.18 | 0.09 |
| ConPS+Gurobi | 97286.74 | 88.34 | 125.13 | 0.18 | 11.34 | 0.13 | 703.17 | 0.08 |
| HiPO+Gurobi | **97354.78** | **20.30** | **125.04** | **0.09** | **11.24** | **0.03** | **703.13** | **0.04** |
| Improvement | | 84.1% | | 80.9% | | 72.7% | | 77.8% |
| SCIP | 96049.43 | 1325.65 | 127.31 | 2.36 | 18.00 | 6.79 | 706.05 | 2.96 |
| PS+SCIP | 96212.73 | 1162.35 | 126.76 | 1.81 | 17.51 | 6.30 | 705.77 | 2.68 |
| ConPS+SCIP | 96294.93 | 1080.15 | 126.54 | 1.59 | 17.08 | 5.87 | 705.68 | 2.59 |
| HiPO+SCIP | **96354.57** | **1020.51** | **126.02** | **1.07** | **16.54** | **5.33** | **705.63** | **2.54** |
| Improvement | | 23.0% | | 54.7% | | 21.5% | | 14.2% |

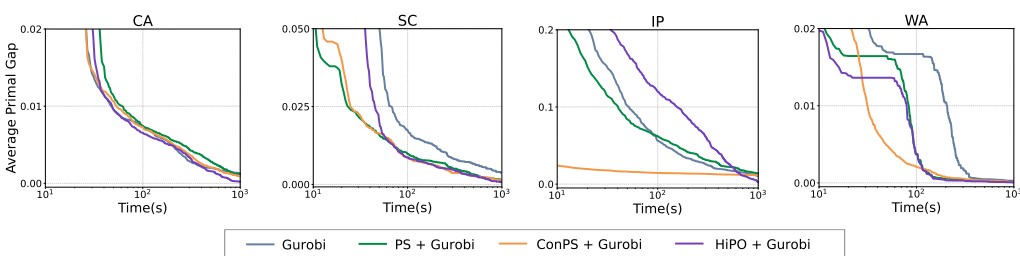

Figure 3: The primal gap of the approaches as the solving process proceeds. Our methods are implemented using Gurobi, with a time limit set to 1,000s, and we average the results across 100 testing instances. A lower primal gap for our method indicates stronger convergence performance.

performs PS across most datasets, which reflects that incorporating solution information of varying quality facilitates more effective model learning. The results demonstrate that HiPO-MILP outperforms all baselines uniformly and significantly: it achieves the best Obj values and the lowest $gap_{abs}$ across all benchmark datasets. Specifically, HiPO-MILP reduces $gap_{abs}$ substantially: it delivers up to an 84.1% improvement in $gap_{abs}$ reduction compared to Gurobi on the CA benchmark, and a 54.7% improvement compared to SCIP on the SC benchmark. For the more complex IP and WA benchmarks, HiPO-MILP also maintains advantages in both Obj optimality and $gap_{abs}$ reduction. In addition, we also conduct real-world datasets from MIPLIB and report the results in Appendix F.1.

**Primal Gap as a Function of Runtime** Figure 3 illustrates the curves of the relative primal gap over the solving process, where the relative primal gap is defined as $gap_{rel} := |OBJ - BKS|/|BKS|$. As shown in Figure 3, while the initial reduction in the gap appears less steep, this observation is primarily a result of the x-axis scaling. In practice, HiPO-MILP can achieve high-quality solutions within the first 100 seconds of the solving process. This performance advantage stems from the more precise variable selection in our proposed approach: the fixed variables predicted by HiPO-MILP provide a superior initial starting point for the solver, thereby enabling more efficient local search. In contrast, baseline methods frequently make hasty variable selection decisions based solely on pointwise training losses. This not only leads to misidentification of critical variables but also causes premature convergence to suboptimal solutions. Furthermore, HiPO-MILP consistently attains better final objective values, which is an outcome that further confirms its inherent advantages in both convergence speed and solution quality.

## 5.3 ABLATION STUDY

**Preference Hierarchy** We conduct an ablation study to examine how the three-tiered preference scheme contributes to HiPO-MILP's performance. We implement three variants: one omitting the top tier $S^*$, one omitting the middle tier $S_2$, and one omitting the lowest tier $S_3$. Table 2 reports the average best objective values and absolute primal gaps under a 1,000s time limit. The results show that removing any tier degrades performance, but omitting the highest tier $S^*$ causes the most significant drop in quality and increase in gap. This indicates that the top preference layer plays a critical guiding role in steering the solver toward high-quality regions. And the other preference hierarchies could help the model learn a deeper understanding of MILP instances.

Table 2: Comparison of solving performance of HiPO-MILP with different preference strategies, under a 1,000s time limit. We report the average best objective values and absolute primal gap. '↑' denotes that higher is better, while '↓' indicates that lower is better. We mark the **best values** in bold.

| Methods | CA (BKS 97375.08) | | SC (BKS 124.95) | |
|---|---|---|---|---|
| | Obj ↑ | $\text{gap}_{abs}$ ↓ | Obj ↓ | $\text{gap}_{abs}$ ↓ |
| HiPO w/o $S^*$+Gurobi | 97264.54 | 110.54 | 125.08 | 0.13 |
| HiPO w/o $S_2$+Gurobi | 97249.06 | 126.02 | 125.11 | 0.16 |
| HiPO w/o $S_3$+Gurobi | 97278.33 | 96.75 | 125.07 | 0.12 |
| HiPO+Gurobi | **97354.78** | **20.30** | **125.04** | **0.09** |

**Preference Pairs** In the preference learning process, for each MILP instance we must sample a number of preference pairs to fine-tune the model. Intuitively, too few preference pairs may starve the model of signal, while too many pairs may introduce noise or redundant comparisons and slow down learning. To assess how the sample size of preference pairs affects overall performance, we evaluate HiPO-MILP using four different sample sizes: 32, 64, 128, and 512 preference pairs per instance. Table 3 records the performance of different sample pairs on CA and SC benchmarks. We see that sampling 64 preference pairs achieves the best result on both benchmarks, increasing to 128 or to 512 leads to slight deterioration, and reducing to 32 also underperforms. This suggests a "sweet spot" in the number of preference comparisons: enough to capture informative contrasts, but not so many as to overconstrain or introduce inconsistent signals.

Table 3: Comparison of solving performance with different sample sizes, under a 1,000s time limit. We report the average best objective values and absolute primal gap. '↑' denotes that higher is better, while '↓' indicates that lower is better. We mark the **best values** in bold.

| Sample Size | CA (BKS 97375.08) | | SC (BKS 124.95) | |
|---|---|---|---|---|
| | Obj ↑ | $\text{gap}_{abs}$ ↓ | Obj ↓ | $\text{gap}_{abs}$ ↓ |
| 32 | 97283.51 | 91.57 | 125.06 | 0.11 |
| 64 | **97354.78** | **20.30** | **125.04** | **0.09** |
| 128 | 97350.07 | 25.01 | 125.09 | 0.14 |
| 512 | 97273.55 | 101.53 | 125.07 | 0.12 |

In addition, we also conduct a more detailed analysis of the hyperparameters in preference learning and search, and further explore the impacts on model performance of two key factors, namely the number of training instances and top near-optimal solutions which served as labels. All these results are presented in Appendix F.2.

## 6 CONCLUSION AND FUTURE WORKS

In this paper, we present a new framework called HiPO-MILP that uses preference learning to fully exploit information across solutions of varying quality. We define a holistic quality score that combines the objective value and solution distance to assess solution quality. Based on this score, we organize solutions into a three-tier hierarchical preference structure and use these preferences to improve the efficiency of extracting information from solutions. Experimental results show that HiPO-MILP significantly outperforms other ML-based methods in terms of solution quality and demonstrates strong generalization ability and promising potential for real-world application.

ETHICS STATEMENT.

This work is designed to explore the significance of the solution prediction methods in solving mixed-integer linear programming problems. We do not foresee any direct, immediate, or negative societal impacts of our research.

REPRODUCIBILITY STATEMENT.

All the results in this work are reproducible. We have discussed the implementation details in Section 5.1 and Appendix B, including the data generation process in Appendix B.1, the training details in Appendix B.2, and inference details in Appendix B.3.

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

## USE OF LLMS

Large language models (LLMs) were used to aid writing polish, including refining sentence phrasing, logical flow, and prose clarity, without altering original meanings or technical details. LLMs did not participate in core research tasks (e.g., experiment design, data processing, model training, result analysis, or drafting key technical content).

## A    PROOF

### A.1    THEOREM 4.1

*Proof.* Let $\boldsymbol{y} \in \text{conv}(S^*)$ be a feasible solution to the MILP. We aim to show $\boldsymbol{c}^\top \boldsymbol{y} = z^*$.

By the definition of the convex hull, $\boldsymbol{y}$ can be written as a convex combination of elements in $S^*$:

$$\boldsymbol{y} = \sum_{i=1}^k \lambda_i \boldsymbol{x}^{(i)} \quad \text{with } \boldsymbol{x}^{(i)} \in S^*, \ \lambda_i \geq 0, \ \sum_{i=1}^k \lambda_i = 1.$$

The objective value of $\boldsymbol{y}$ is computed using the linearity of the objective function $\boldsymbol{c}^\top \boldsymbol{x}$:

$$\boldsymbol{c}^\top \boldsymbol{y} = \boldsymbol{c}^\top \left( \sum_{i=1}^k \lambda_i \boldsymbol{x}^{(i)} \right) = \sum_{i=1}^k \lambda_i \left( \boldsymbol{c}^\top \boldsymbol{x}^{(i)} \right).$$

By definition of $S^*$, every $\boldsymbol{x}^{(i)} \in S^*$ achieves the optimal objective value: $\boldsymbol{c}^\top \boldsymbol{x}^{(i)} = z^*$ for all $i$. Substituting this into the equation above gives:

$$\boldsymbol{c}^\top \boldsymbol{y} = \sum_{i=1}^k \lambda_i z^* = z^* \cdot \sum_{i=1}^k \lambda_i.$$

Since $\sum_{i=1}^k \lambda_i = 1$ (by the definition of a convex combination), we conclude:

$$\boldsymbol{c}^\top \boldsymbol{y} = z^*.$$

Thus, any feasible solution within the convex hull of the optimal set $S^*$ must have the optimal objective value $z^*$.    □

### A.2    THEOREM 4.2

*Proof.* For a compact convex set $C \subseteq \mathbb{R}^n$ and a point $\boldsymbol{y} \in \mathbb{R}^n$, the distance minimization problem

$$\min_{\boldsymbol{x} \in C} \|\boldsymbol{y} - \boldsymbol{x}\|$$

has a unique solution when using the Euclidean norm. By compactness of $C$, the continuous function $f(\boldsymbol{x}) = \|\boldsymbol{y} - \boldsymbol{x}\|$ attains its infimum on $C$ (Extreme Value Theorem). Thus, there exists at least one $\boldsymbol{x}^* \in C$ such that $\|\boldsymbol{y} - \boldsymbol{x}^*\| = d(\boldsymbol{y}, C)$. Suppose, for contradiction, there exist two distinct points $\boldsymbol{x}^{(1)}, \boldsymbol{x}^{(2)} \in C$ such that

$$\|\boldsymbol{y} - \boldsymbol{x}^{(1)}\| = \|\boldsymbol{y} - \boldsymbol{x}^{(2)}\| = d(\boldsymbol{y}, C).$$

By convexity of $C$, the midpoint $\boldsymbol{x}^{(m)} = \frac{1}{2}\boldsymbol{x}^{(1)} + \frac{1}{2}\boldsymbol{x}^{(2)} \in C$. By the strict convexity of the Euclidean norm:

$$\|\boldsymbol{y} - \boldsymbol{x}^{(m)}\| < \frac{1}{2}\|\boldsymbol{y} - \boldsymbol{x}^{(1)}\| + \frac{1}{2}\|\boldsymbol{y} - \boldsymbol{x}^{(2)}\| = d(\boldsymbol{y}, C),$$

which contradicts the definition of $d(\boldsymbol{y}, C)$ as the infimum. Thus, $\boldsymbol{x}^*$ must be unique.    □

### A.3    THEOREM 4.3

*Proof.* A solution $\boldsymbol{y} \in S \setminus S^*$ (non-near-optimal) fails to satisfy at least one condition of $S^*$. We analyze both failure cases.

**Case 1:** $y \notin S^*$ **because** $c^\top y > z^* + \epsilon_1$   For $x \in S^*$, by definition of $S^*$: 1. Objective term: $c^\top x \leq z^* + \epsilon_1$. Since $c^\top y > z^* + \epsilon_1$, we have $c^\top x \leq c^\top y$. 2. Distance term: Even if $d(x, \mathrm{conv}(S^*)) = d(y, \mathrm{conv}(S^*))$, the smaller $c^\top x$ implies:

$$\frac{c^\top x}{\tau_{\mathrm{obj}}} \leq \frac{c^\top y}{\tau_{\mathrm{obj}}}$$

Combine the two terms in $Q(\cdot)$:

$$\frac{c^\top x}{\tau_{\mathrm{obj}}} + \lambda \cdot \frac{d(x, \mathrm{conv}(S^*))}{\tau_{\mathrm{dist}}} \leq \frac{c^\top y}{\tau_{\mathrm{obj}}} + \lambda \cdot \frac{d(y, \mathrm{conv}(S^*))}{\tau_{\mathrm{dist}}}$$

Since $Q(\cdot)$ is the negative of the above sum, taking the negative reverses the inequality:

$$Q(x) \geq Q(y)$$

**Case 2:** $y \notin S^*$ **because** $d(y, \mathrm{conv}(S^*)) > \epsilon_2$   For $x \in S^*$, by definition of $S^*$: 1. Distance term: $d(x, \mathrm{conv}(S^*)) \leq \epsilon_2$. Since $d(y, \mathrm{conv}(S^*)) > \epsilon_2$, we have $d(x, \mathrm{conv}(S^*)) \leq d(y, \mathrm{conv}(S^*))$. 2. Objective term: Even if $c^\top x = c^\top y$, the smaller distance implies:

$$\lambda \cdot \frac{d(x, \mathrm{conv}(S^*))}{\tau_{\mathrm{dist}}} \leq \lambda \cdot \frac{d(y, \mathrm{conv}(S^*))}{\tau_{\mathrm{dist}}}$$

Combine the two terms in $Q(\cdot)$:

$$\frac{c^\top x}{\tau_{\mathrm{obj}}} + \lambda \cdot \frac{d(x, \mathrm{conv}(S^*))}{\tau_{\mathrm{dist}}} \leq \frac{c^\top y}{\tau_{\mathrm{obj}}} + \lambda \cdot \frac{d(y, \mathrm{conv}(S^*))}{\tau_{\mathrm{dist}}}$$

Taking the negative (per $Q(\cdot)$ definition) reverses the inequality:

$$Q(x) \geq Q(y)$$

$\square$

# B   IMPLEMENTATION DETAILS

Recent advances in machine learning and graph neural networks have enabled a wave of ML-augmented approaches for MILP solving (Nair et al., 2020; Han et al., 2023; Huang et al., 2024; Liu et al., 2025; Geng et al., 2025), with strong results on challenging benchmarks. In our implementation, we follow the PS (Han et al., 2023) framework as described in the original paper and released code. HiPO-MILP is finetuned on top of PS and uses the same backbone, a graph neural network with four half-convolution layers. Because an official implementation of ConPS (Huang et al., 2024) has not been released, we re-implemented it and carefully tuned hyperparameters to obtain competitive performance. All experiments were run on a single workstation with an NVIDIA GeForce RTX 3090 GPU and an AMD EPYC 7402 24-core CPU.

## B.1   DATA GENERATION

Following the experimental setup in Han et al. (2023), we employ 240 instances for training, 60 instances for validation, and 100 instances for testing. All reported results reflect the average performance across the 100 test instances. During data generation, we compute the convex hull over a set of near-optimal solutions and construct perturbed solutions, enabling us to derive the holistic quality score for each hierarchical preference solution.

**Convex hull computation**   We start by collecting a batch of high-quality solutions for each problem instance. Intuitively, the quality and diversity of these solutions can strongly influence the effectiveness of model learning. In Appendix F.2, we further analyze how the number of near-optimal solutions used as labels affects learning. Rather than solely weighting these solutions by their objective values for supervision, we also incorporate pairwise distance information among them. To do so, we extract those solutions whose objective value is equal to the best objective value among the collected set, and treat them as the vertices of a convex hull. Formally, let $\{x^{(i)}\}_{i=1}^K \subset \{0,1\}^n$

denote the $K$ "best" solutions sharing the same objective $f^* = \min_i c^\top x^{(i)}$. For any candidate solution $x$, we project it onto the convex hull spanned by $\{x^{(i)}\}$ by solving

$$\min_{\lambda \in \mathbb{R}^K} \|V\lambda - x\|_2^2 \quad \text{subject to} \quad \lambda \succeq 0, \ \mathbf{1}^\top \lambda = 1, \tag{10}$$

where $V = [x^{(1)}, x^{(2)}, \ldots, x^{(K)}] \in \mathbb{R}^{n \times K}$. We solve this convex Quadratic Programming (QP) (regularized by a small diagonal term $\varepsilon I$) using Operator Splitting Quadratic Program (OSQP). After obtaining $\lambda$, we compute $x_{\text{proj}} = V\lambda$ and define the distance

$$d(x) = \|x_{\text{proj}} - x\|_2. \tag{11}$$

If all solvers fail or the QP is ill-posed, we approximate distance by Hamming distance to the nearest vertex, i.e.

$$d(x) \approx \min_{i=1,\ldots,K} \|x - x^{(i)}\|_1 \tag{12}$$

over the top-$k$ vertices.

**Perturbed solution construction** We adopt a perturbation strategy inspired by the ConPS Huang et al. (2024) to generate perturbed solutions. There are two types of perturbed solutions: infeasible perturbations and low-quality feasible perturbations. For infeasible perturbations, for each high-quality solution, we randomly flip a fixed proportion (initially 10 %) of its binary variables. We then fix those perturbed binary assignments and check if the resulting subproblem becomes infeasible (for mixed integer programs we verify feasibility over continuous variables). If we fail to generate enough infeasible candidates, we gradually increase the flip rate by 5% until sufficient samples are collected. For low-quality perturbations we aim to find feasible solutions that are similar to the positive sample yet exhibit worse objective values. In the pure binary case we solve a local branching subproblem that maximizes $c^\top x'$ under the constraint that the Hamming distance to the positive sample does not exceed a threshold (e.g. 10 units). We then filter solutions whose objective difference exceeds a preset margin. In the mixed integer case we treat the problem as a max–min optimization: we fix the binary part within a small distance, then optimize continuous variables to maximize the worst objective value. We iteratively impose constraints that force the next solution to be worse than the previous, thereby extracting degraded feasible solutions. After obtaining these perturbed solutions, we compute their holistic quality scores in the same fashion as for the other solutions.

## B.2 TRAINING DETAILS

During model training, we adopt SimPO for preference learning to better distinguish among different solutions. We observe that in many solution pairs, the majority of decision variables share identical values. If we naively define the preference score as

$$r_\theta(\mathbf{x}) = \beta \log \pi_\theta(\mathbf{x} \mid \mathcal{I}) = \beta \sum_{j=1}^p \log \pi_\theta(x_j \mid \mathcal{I}), \tag{13}$$

then the contributions of the few differing variables can be diluted by the many invariant ones. To mitigate this issue, in practice we focus only on the variables that differ between the two solutions. Concretely, for a pair $(\mathbf{x}^+, \mathbf{x}^-)$, let $\Delta = \{j \mid x_j^+ \neq x_j^-\}$. We then define a corrected preference score

$$r_\theta(\mathbf{x}^+, \mathbf{x}^-) = \beta \sum_{j \in \Delta} \left(\log \pi_\theta(x_j^+ \mid \mathcal{I}) - \log \pi_\theta(x_j^- \mid \mathcal{I})\right). \tag{14}$$

This adjustment ensures that the learning signal concentrates on the truly discriminative coordinates.

Moreover, when sampling preference pairs, we impose a threshold on the difference between the positive and negative samples. If two solutions are extremely similar, forcing a preference label may lead to overfitting. Therefore, we require that sampled pairs differ by at least a minimal margin both in objective value and in solution distance. This constraint guarantees that preference supervision remains meaningful. In Appendix F.2 we provide detailed analysis of how these thresholds affect model performance.

### B.3 INFERENCE DETAILS

In the PS framework, the hyperparameters $k_0$, $k_1$, and $\delta$ play a critical role in balancing search efficiency and solution quality. Because our benchmarks are more challenging than those in the original PS paper, the PS hyperparameters reported therein yield poor performance in our experiments. Therefore, we carefully re-tune these search parameters, and we enforce that HiPO-MILP uses exactly the same search parameters as PS, to rule out disparities from differing hyperparameter settings. Table 4 summarizes the parameter choices used for all methods in this work, and in Appendix F.2 we include a full ablation study of these hyperparameters.

Table 4: The partial solution size parameter $(k_0, k_1)$ and neighborhood parameter $\Delta$.

| Benchmark | CA | SC | IP | WA |
|---|---|---|---|---|
| PS+Gurobi | (900,0,80) | (4900,40,30) | (60,35,55) | (200,600,100) |
| ConPS+Gurobi | (900,0,50) | (1000,0,200) | (400,5,3) | (0,500,10) |
| HiPO+Gurobi | (900,0,80) | (4900,40,30) | (60,35,55) | (200,600,100) |
| PS+SCIP | (900,0,80) | (4900,40,30) | (60,35,55) | (200,600,100) |
| ConPS+SCIP | (900,0,50) | (1000,0,200) | (400,5,3) | (0,400,50) |
| HiPO+SCIP | (900,0,80) | (4900,40,30) | (60,35,55) | (200,600,100) |

## C DETAILS ON BENCHMARKS

### C.1 BENCHMARKS IN MAIN EVALUATION

In our primary evaluation we rely on four categories of MILP benchmarks. The Combinatorial Auction (CA) and Set Cover (SC) instances are generated via the procedures introduced in Gasse et al. (2019). In particular, the CA family follows the auction-generation scheme from Leyton-Brown et al. (2000), while the SC instances are produced based on the classical set-cover generation method in Balas & Ho (1980). The IP and WA datasets are sourced from the NeurIPS ML4CO 2021 challenge (Gasse et al., 2022). Table 5 reports key statistics (number of constraints, variables, binary/continuous splits) for all these instance classes used in our experiments.

Table 5: Statistical information of the benchmarks we used in this paper.

| | CA | SC | IP | WA |
|---|---|---|---|---|
| # Constraints | 2593 | 3000 | 195 | 64306 |
| # Variables | 1500 | 5000 | 1083 | 61000 |
| # Binary Variables | 1500 | 5000 | 1050 | 1000 |
| # Continuous Variables | 0 | 0 | 33 | 60000 |
| # Integer Variables | 0 | 0 | 0 | 0 |

### C.2 BENCHMARKS IN USED FOR GENERALIZATION

To assess the generalization capability of our methods beyond the training scale, we synthesize larger versions of the CA and SC problem families. Using the same generation code as Gasse et al. (2019), we produce CA instances with around 2,596 constraints and 4,000 variables on average, and SC instances with approximately 6,000 constraints and 10,000 variables. These enlarged instances

are substantially more demanding than those in our training set, and thus serve as a stress test for cross-scale generalization.

## C.3 SUBSET OF MIPLIB

To validate HiPO-MILP on realistic combinatorial challenges, we assemble a targeted benchmark derived from the MIPLIB 2017 collection (Gleixner et al., 2021). MIPLIB offers a rich variety of mixed-integer programs drawn from real applications in areas like logistics, energy, network design, and scheduling. Because the full collection is highly heterogeneous and often too large for learning-based models to train on end to end, we select a manageable yet representative subset. Our selection follows an instance similarity approach inspired by prior methods (e.g. Apollo-MILP (Liu et al., 2025)). Concretely, similarity is measured using one hundred pre-defined instance features (covering constraint structure, variable statistics, objective coefficients, etc.) as introduced in the MIPLIB documentation (Gleixner et al., 2021). We begin with the "IIS" family—there are 11 candidate instances in total. From these, we allocate six for training (namely `glass-sc`, `iis-glass-cov`, `5375`, `214`, `56133`, and `iis-hc-cov`) and reserve the remaining five (i.e. `ex1010-pi`, `fast0507`, `ramos3`, `scpj4scip`, and `scpl4`) for testing. To increase the evaluation challenge, we further include three large and structurally complex instances (`bg512142`, `dws008-01`, `dws008-03`), which feature more variables, more constraints, and richer variable/constraint types. Table 6 collects the detailed statistics of all test instances in our MIPLIB benchmark.

Table 6: Statistical information of the instances in the constructed MIPLIB dataset.

| | # Constraints | # Variables | # Binaries | # Integers | # Continuous | # Nonzero Coefficient |
|---|---|---|---|---|---|---|
| ex1010-pi | 1468 | 25200 | 25200 | 0 | 0 | 102114 |
| fast0507 | 507 | 63009 | 63009 | 0 | 0 | 409349 |
| ramos3 | 2187 | 2187 | 2187 | 0 | 0 | 32805 |
| scpj4scip | 1000 | 99947 | 99947 | 0 | 0 | 999893 |
| scpl4 | 2000 | 200000 | 200000 | 0 | 0 | 2000000 |
| bg512142 | 1307 | 792 | 240 | 0 | 552 | 3953 |
| dws008-03 | 16344 | 32280 | 18928 | 0 | 13352 | 165168 |
| dws008-01 | 6064 | 11096 | 6608 | 0 | 4488 | 56400 |

## D HYPERPARAMETERS

Since HiPO-MILP is obtained by fine-tuning the PS model, most training hyperparameters remain identical to those in PS. Table 7 lists the hyperparameters specific to HiPO-MILP.

Table 7: Hyperparameters in our experiments.

| Hyperparameter | CA | SC | IP | WA | Description |
|---|---|---|---|---|---|
| lr | 1e-4 | 1e-5 | 1e-5 | 1e-5 | Learning rate for training. |
| $\beta$ | 0.8 | 0.8 | 0.8 | 0.8 | Temperature parameter in reward. |
| $\gamma$ | 0.2 | 0.2 | 0.2 | 0.2 | Margin in the preference learning. |
| $\eta$ | 5.0 | 0.5 | 0.5 | 0.5 | Balancing the two loss functions. |
| sample numbers | 64 | 32 | 16 | 16 | Sample pairs in each instance. |

## E THE IMPORTANCE OF THE HOLISTIC QUALITY SCORE

To further validate the effectiveness of our holistic quality score, we replace the conventional objective-weighted labels in PS with labels derived from the holistic quality score, train a new model under that scheme, and present the performance comparison in Table 8. We can easily see that the model with the holistic quality score holds superior performance than PS on the four benchmarks.

Table 8: Performance comparison of PS and PS w.i. H.Q.S. H.Q.S denotes the holistic quality score. Two core metrics are reported: Objective value and absolute gap. '↑' denotes higher is better, while '↓' indicates lower is better. **Best values** are marked in bold. "Improvement" quantifies the gap optimization percentage of PS w.i. H.Q.S.+Gurobi over PS+Gurobi.

| Method | CA (BKS: 97375.08) | | SC (BKS: 124.95) | | IP (BKS: 11.21) | | WA (BKS: 703.09) | |
|---|---|---|---|---|---|---|---|---|
| | Obj ↑ | $\text{gap}_{abs}$ ↓ | Obj ↓ | $\text{gap}_{abs}$ ↓ | Obj ↓ | $\text{gap}_{abs}$ ↓ | Obj ↓ | $\text{gap}_{abs}$ ↓ |
| PS+Gurobi | 97257.10 | 117.98 | 125.11 | 0.16 | 11.37 | 0.16 | 703.18 | 0.09 |
| PS w.i H.Q.S.+Gurobi | **97350.97** | **24.11** | **125.02** | **0.07** | **11.31** | **0.10** | 703.15 | **0.06** |
| Improvement | | 79.6% | | 56.3% | | 37.5% | | 33.3% |

# F  ADDITIONAL EXPERIMENTAL RESULTS

## F.1  REAL-WORLD DATASET

To demonstrate HiPO-MILP's practicality on real-world instances, we test it over a curated subset of MIPLIB problems. For PS and ConPS, we train the model on the training dataset which are selected based on the instance similarity. In contrast, as a fine-tuning approach, HiPO-MILP exploits detailed information about solutions: during dataset preparation, we collect 500 high-quality feasible solutions per instance and use these to fine-tune the model that was pretrained on the synthetic dataset. At inference time, we hold the search hyperparameters constant across all runs, $k_0 = 0.6$, $k_1 = 0.01$, and $\Delta = 1000$, rather than adjusting them per instance. This consistency underscores the robustness of HiPO-MILP. Table 9 reports the performance on the MIPLIB subset. HiPO-MILP either matches or surpasses baseline methods in nearly all cases, and interestingly it attains the best known solution (BKS) on all five test instances on which the baselines fail. These findings strongly support that HiPO-MILP sustains excellent performance even on challenging real-world MILP problems.

Table 9: The best objectives found by the approaches on each test instance in MIPLIB. *BKS* represents the best objectives from the website of MIPLIB.

| | BKS | Gurobi | PS+Gurobi | ConPS+Gurobi | HiPO-MILP+Gurobi |
|---|---|---|---|---|---|
| ex1010-pi | 233.00 | 239.00 | 241.00 | 239.00 | **237.00** |
| fast0507 | 174.00 | 174.00 | 179.00 | 179.00 | **174.00** |
| ramos3 | 186.00 | 233.00 | 225.00 | 225.00 | **224.00** |
| scpj4scip | 128.00 | 132.00 | 133.00 | 133.00 | **131.00** |
| scpl4 | 259.00 | 277.00 | 275.00 | 275.00 | **273.00** |
| dws008-03 | 62831.76 | 64452.67 | 71234.06 | 67473.85 | **65685.03** |
| dws008-01 | 37412.60 | 37412.60 | 39043.26 | 38817.50 | **37412.6** |
| bg512142 | 184202.75 | 184202.75 | 190193.00 | 190193.00 | **188634.5** |

## F.2  HYPERPARAMETER ANALYSIS

Table 10: Effect of temperature $\beta$ on CA benchmark.

| | Obj ↑ | $\text{gap}_{abs}$ ↓ |
|---|---|---|
| $\beta = 0.2$ | 97284.92 | 90.16 |
| $\beta = 0.5$ | 97338.25 | 36.83 |
| $\beta = 0.8$ | **97354.78** | **20.30** |
| $\beta = 1.0$ | 97274.81 | 100.27 |

**Analysis of temperature** $\beta$  In the preference learning, the temperature (inverse scale $\beta$) governs how sharply the model's preference (or reward) distribution emphasizes differences in predicted scores: a higher $\beta$ (lower "temperature") makes the model more confident in selecting high-scoring items, while a lower $\beta$ softens the distinctions and encourages exploration. In the MILP setting, this

tradeoff becomes more delicate, because the combinatorial solution space is huge and true optimal solutions tend to lie at extreme corners: if $\beta$ is too low, the model may fail to discriminate among many feasible solutions, whereas if $\beta$ is too high, early mis-preferences risk biasing the search irreversibly. Table 10 reports the model performance of different $\beta$ values on CA benchmark.

**Analysis of margin $\gamma$** In preference-learning settings, the margin $\gamma$ determines how strongly a preferred item must exceed a dispreferred item in score to enforce a confident preference, effectively controlling how "aggressive" the preference separation is. In the context of HiPO-MILP, choosing a large $\gamma$ biases the model to strictly favor higher-scoring candidates and may risk overconfidence or rigidity, while a small $\gamma$ yields softer separations and more uncertainty in preference ranking; thus, $\gamma$ modulates the tradeoff between discriminative power and robustness in HiPO-MILP's preference modeling. Table 11 reports the model performance of different $\gamma$ values on CA benchmark.

Table 11: Effect of margin $\gamma$ on CA benchmark.

|  | Obj ↑ | $\text{gap}_{abs}$ ↓ |
|---|---|---|
| $\gamma = 0.1$ | 97118.51 | 256.57 |
| $\gamma = 0.2$ | **97354.78** | **20.30** |
| $\gamma = 0.5$ | 97195.32 | 179.76 |
| $\gamma = 1.0$ | 97068.19 | 306.89 |

**Analysis of pair interval** The way we sample preference pairs profoundly influences how strong and clear the learning signal becomes. If a positive and a negative example differ only trivially, the model learns very little; if they differ ambiguously, noise drowns out meaningful signal. In HiPO-MILP, the notion of a "pair interval" is central, because it governs how bold or conservative the model's preference judgments are. A wide interval causes the model to focus only on well-separated pairs, enforcing strong separation but potentially missing subtler distinctions; a narrow interval allows many pairs but risks admitting weak or conflicting signals, which degrades learning speed and reliability. It is essential to make a distinction between two kinds of thresholds: in the objective domain, the interval defines how large a score difference must be for a pair to count as informative; in the solution domain, it defines how dissimilar two candidate solutions must be before we treat them as meaningfully comparable. Table 12 shows model performance under different objective-level thresholds on the CA benchmark. And Table 13 shows performance under different minimal distances between solution pairs on the CA benchmark.

Table 12: Model performance for varying objective-level thresholds on the CA benchmark. The thresholds are set as fractions of the span between the maximum and minimum objective values among a solution set obtained from traditional solvers.

|  | Obj ↑ | gap_abs ↓ |
|---|---|---|
| 0.015 | 97265.10 | 109.98 |
| 0.025 | 97309.19 | 65.89 |
| 0.050 | **97354.78** | **20.30** |
| 0.100 | 97298.35 | 76.73 |

Table 13: Model performance across different solution-level distances between paired solutions on the CA benchmark. We enforce that each pair differ in at least a minimum number of variable assignments.

|  | Obj ↑ | $\text{gap}_{abs}$ ↓ |
|---|---|---|
| 3 | 97321.58 | 53.50 |
| 5 | 97247.71 | 127.37 |
| 8 | **97354.78** | **20.30** |
| 10 | 97290.32 | 84.76 |

**Analysis of $\eta$** The hyperparameter $\eta$ balances between the binary cross-entropy loss and the HiPO-MILP preference loss: a larger $\eta$ gives more weight to BCE and thus stabilizes classification,

whereas a smaller $\eta$ emphasizes the HiPO-MILP loss and strengthens preference learning. Table 14 reports the model performance of different $\eta$ values between pair solutions on CA and SC benchmarks.

Table 14: Effect of $\eta$ on performance in CA and SC benchmarks.

| | CA (BKS: 97375.08) | | SC (BKS: 124.95) | |
|---|---|---|---|---|
| | Obj $\uparrow$ | gap$_{abs}$ $\downarrow$ | Obj $\downarrow$ | gap$_{abs}$ $\downarrow$ |
| $\eta = 1.0$ | 97272.19 | 102.89 | **125.04** | textbf0.09 |
| $\eta = 3.0$ | 97332.98 | 42.10 | 125.16 | 0.21 |
| $\eta = 5.0$ | **97354.78** | **20.30** | 125.08 | 013 |
| $\eta = 8.0$ | 97345.70 | 29.38 | 125.24 | 0.29 |

**Analysis of training instance numbers** We investigate how reducing the size of the training set affects the performances of PS and HiPO-MILP. Since HiPO-MILP more fully exploits preference relationships inferred from data, we expect its performance to degrade more gracefully under data scarcity, whereas PS may suffer greater decline as it depends more directly on raw samples. Table 15 reports model performance with different training data.

Table 15: Model performance under different numbers of training instances on the CA benchmark. "80 %", "50 %" and "30 %" indicate that only those proportions of the original training set are used.

| | 100% | | 80% | | 50% | | 30% | |
|---|---|---|---|---|---|---|---|---|
| Method | Obj $\uparrow$ | gap$_{abs}$ $\downarrow$ | Obj $\uparrow$ | gap$_{abs}$ $\downarrow$ | Obj $\uparrow$ | gap$_{abs}$ $\downarrow$ | Obj $\uparrow$ | gap$_{abs}$ $\downarrow$ |
| PS+Gurobi | 97257.10 | 117.98 | 97158.02 | 217.06 | 97074.86 | 300.22 | 97003.88 | 371.20 |
| HiPO+Gurobi | **97354.78** | **20.30** | **97290.64** | **84.44** | **97228.99** | **146.09** | **97167.44** | **207.64** |

**Analysis of the number of top near-optimal solutions** In the PS paradigm guided only by objective values, relying on a small top-$k$ set risks overfitting: some solutions may have excellent objective scores yet lie far from the true optimal region in solution space, leading the model toward misleading local optima. HiPO-MILP, in contrast, alleviates this issue by simultaneously considering both objective values and solution distances, thus preventing undue trust in structurally distant but "objectively good" solutions. Furthermore, HiPO-MILP does not treat the top solutions as hard labels; instead it learns fine-grained preference relations among them, allowing more complete use of the information embedded in a larger solution set and making its performance less sensitive to the precise choice of $k$. Based on this insight, we conducted experiments varying $k$ (e.g. top 5, 10, 20) to compare the sensitivity of PS and HiPO-MILP to $k$, and we report the results in Table 16.

Table 16: Model performance with varying numbers of near-optimal solutions used as training labels on the CA benchmark. "H.Q.S." denotes the holistic quality score, and "PS w. H.Q.S." denotes PS where the weighted solutions are replaced by objective values as the holistic quality score.

| | 10 | | 20 | | 50 | |
|---|---|---|---|---|---|---|
| Method | Obj $\uparrow$ | gap$_{abs}$ $\downarrow$ | Obj $\uparrow$ | gap$_{abs}$ $\downarrow$ | Obj $\uparrow$ | gap_abs $\downarrow$ |
| PS+Gurobi | 97257.10 | 117.98 | 97126.06 | 249.02 | 97177.60 | 197.48 |
| PS wi. H.Q.S+Gurobi | 97350.97 | 24.11 | **97306.73** | **68.35** | 97315.97 | 59.11 |
| HiPO+Gurobi | **97354.78** | **20.30** | 97293.51 | 81.57 | **97319.29** | **55.79** |

**Sensitivity analysis of $k_0$, $k_1$, and $\Delta$** Because we are working on a dataset with far higher scale in variables and constraints than those typically used by PS, the original PS search parameters perform quite poorly. We therefore made a careful effort to fine-tune them to find reasonably good settings. To avoid any bias in comparing HiPO-MILP to PS, in all experiments we fix PS's tuned parameters and use the same settings for HiPO-MILP. Figure 4 shows the performance of PS and HiPO-MILP on the CA dataset under varying values of $k_0$, $k_1$, and $\Delta$. Overall, across almost all parameter configurations, HiPO-MILP considerably outperforms PS.

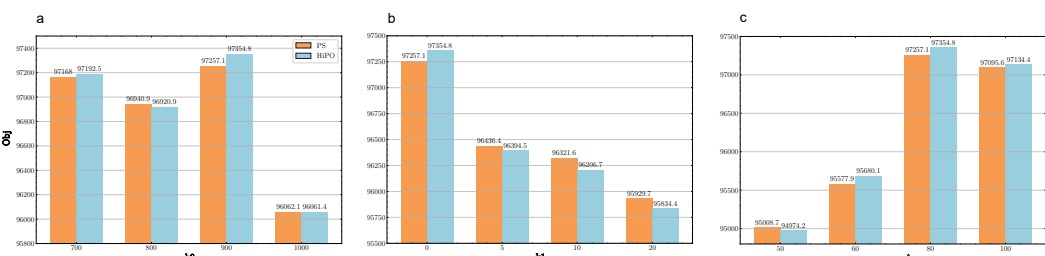

Figure 4: Sensitivity analysis of $k_0$, $k_1$, and $\Delta$ on the CA benchmark. The default configuration $(k_0, k_1, \Delta)$ is $(900, 0, 80)$. (a)–(c) show performance when varying $k_0$, $k_1$, and $\Delta$ respectively while holding the other two parameters fixed.

### F.3 GENERALIZATION

To further assess the generalization power of HiPO-MILP, we evaluate its performance on expanded CA and SC instances with much larger numbers of constraints and variables, as detailed in Appendix C.2. We apply the HiPO-MILP model trained on the primary benchmark directly, without any additional finetuning, to these scaled-up instances. Table 17 summarizes the outcomes, confirming that HiPO-MILP can successfully generalize to more demanding and structurally complex problems.

Table 17: Generalization results of 100 larger instances on CA and SC using Gurobi with a 1000 seconds time limit. '↑' indicates that higher is better, and '↓' indicates that lower is better. We mark the **best values** in bold.

| Method | CA (BKS: 115498.51) | | SC (BKS: 102.09) | |
| --- | --- | --- | --- | --- |
| | Obj ↑ | gap$_{abs}$ ↓ | Obj ↓ | gap$_{abs}$ ↓ |
| Gurobi | 114672.25 | 826.26 | 103.85 | 1.76 |
| PS+Gurobi | 114940.13 | 558.38 | 103.93 | 1.84 |
| ConPS+Gurobi | 115053.87 | 444.64 | 103.84 | 1.75 |
| HiPO-MILP+Gurobi | **115154.14** | **344.37** | **102.93** | **0.84** |

## G DISCUSSIONS

### G.1 LIMITATIONS

While HiPO-MILP demonstrates significant improvements in MILP solving efficiency through hierarchical preference optimization, it still faces certain limitations. One primary constraint lies in the dependency on the quality and diversity of the preference pairs sampled during training; although our method constructs a three-tiered hierarchy to enhance the learning signal, sampling sufficiently informative and high-quality pairs remains challenging, especially for highly complex or degenerate MILP instances where the distinction between near-optimal and perturbed solutions may be subtle. Future work could explore more efficient methods for generating preference pairs or adaptive strategies for tuning these parameters dynamically. Moreover, exploring alternative preference optimization algorithms beyond SimPO, or incorporating reinforcement learning to dynamically adjust the preference hierarchy during training, might lead to more robust and sample-efficient learning.

### G.2 FUTURE AVENUES

There are several promising directions for extending HiPO-MILP. First, the framework could be adapted and fine-tuned for more complex, real-world MILP applications, such as large-scale supply chain optimization, energy grid management, or financial portfolio planning, where instance structures are highly heterogeneous and often contain domain-specific constraints; by leveraging transfer learning or domain adaptation techniques, HiPO-MILP could be tailored to these settings, potentially offering even greater performance gains. Second, further advancements could be made in

the design of the underlying model architecture——for instance, integrating more expressive graph neural networks or attention-based mechanisms to better capture the intricate relationships between variables and constraints in MILP instances.

