# OpenReview forum: "HiPO-MILP: Hierarchical Preference Optimization for MILP Solving"
_ICLR.cc/2026/Conference — ICLR 2026 Conference Withdrawn Submission_

### Official Review · Reviewer_zEPs · 2025-10-29

**Soundness:** 2
**Presentation:** 3
**Contribution:** 2
**Rating:** 2
**Confidence:** 4

**Summary:**

This paper introduces HiPO-MILP, an extension of the predict-and-search (PS) framework that incorporates two key advancements: (i) using the distance to the optimal region as a scoring criterion for candidate solutions, and (ii) employing a hierarchical preference-based loss function. Experiments on widely-used public benchmarks demonstrate its superiority over both PS and contrastive PS.

**Strengths:**

The concept of a holistic quality score, which integrates geometric and algebraic properties, offers a more comprehensive evaluation of candidate solutions and is a noteworthy contribution.

**Weaknesses:**

1. **Insufficient baselines**: Section 2.1 acknowledges a wide range of learning-to-optimize methods. Under the "learn to accelerate" paradigm, learning-based LNS approaches—such as CL-LNS [1] and BTBS-LNS [2]—are promising and should be included for comparison. Similarly, under the "end-to-end learning" paradigm, methods like DiffILO and Apollo-MILP are mentioned but not evaluated. **The primary goal is to enhance MILP solving using any technique, rather than improving a particular pipeline (e.g., PS).**
2. **Insufficient ablation study**: The two proposed improvements—(i) the holistic score and (ii) hierarchical preference optimization—warrant an ablation study to isolate their individual contributions.
 3. **minor comments**: Table 3 indicates that HiPO-MILP is outperformed by ConPS for sample sizes 32 and 512. Does this suggest that HiPO-MILP is sensitive to hyperparameter settings?

[1] Huang, Taoan, et al. "Searching large neighborhoods for integer linear programs with contrastive learning." International conference on machine learning. PMLR, 2023.
[2] Yuan, Hao, et al. "BTBS-LNS: Binarized-Tightening, Branch and Search on Learning LNS Policies for MIP." The Thirteenth International Conference on Learning Representations.

**Questions:**

1.  The authors emphasize the importance of structural proximity to local optima among suboptimal solutions. However, the best solution $x$ obtained in the training set and the optimal solution $x^*$ to the original problem may not be identical. What if $x$ is locally optimal but far from $x^*$? Conversely, suppose a suboptimal solution $x'$ in the training set is close to $x^*$. In that case, $x'$ would be a good candidate but would be penalized under the proposed scoring scheme due to its distance from $x$.
2.  Why is structural proximity considered important? Moreover, Figure 1 does not sufficiently justify this, as linear objectives in MILPs may inherently imply that high-quality solutions lie near the optimal region.
3.  Given the emphasis on structural proximity, if PS and neural diving collect numerous high-quality solutions, why not simply select the best one among them?
4.  In Table 2, when $S_2$ is omitted, it seems like the resulting approach is very similar to contrastive PS? But why does it falls behind ConPS?

---

### Official Review · Reviewer_aJ8X · 2025-10-30

**Soundness:** 1
**Presentation:** 3
**Contribution:** 1
**Rating:** 2
**Confidence:** 4

**Summary:**

This work presents a ML-based approach for MILPs. While the topic is relevant, the paper in its current form has significant shortcomings that prevent it from being a strong contribution. The core motivation is conceptually flawed, the theoretical contributions are overstated, and the empirical validation is insufficient. The method's reliance on multiple hyper-parameters also raises concerns about its practicality and generalizability.

**Strengths:**

The paper is easy to follow.

**Weaknesses:**

### Major

- **Motivation & Figure 1:** The argument that objective value is an inadequate proxy for proximity to optimality is flawed. For a linear objective, the set of optimal solutions lies on a hyperplane. A solution with a near-optimal value is inherently close to this set, making the objective a valid proximity measure. The distance to a single arbitrary optimal point (e.g., $x_3$) is not a meaningful metric.

- **Theoretical Contributions:** The presented "Theorems" (e.g., 4.1-4.3) are definitions or observations, far from novel theoretical results. This overstates the paper's theoretical contribution.

- **Empirical Evaluation:** The scope of the evaluation is limited. Benchmarks on a standard, diverse dataset like **MIPLIB 2017** are necessary to properly assess generalizability and performance.

### Minor Points & Clarifications

- **Hyperparameters:** The method introduces several hyperparameters $(\lambda, \eta, \gamma, \beta)$. A discussion on their sensitivity, tuning cost, and impact on generalization is needed.
- **Unclear Notation:** The symbol $\sigma$ in Eq. (8) is not defined.
- **Grammar:** Use "an MILP" instead of "a MILP".

**Questions:**

See the weaknesses.

---

### Official Review · Reviewer_Coox · 2025-11-01

**Soundness:** 2
**Presentation:** 3
**Contribution:** 2
**Rating:** 4
**Confidence:** 5

**Summary:**

The paper proposes HiPO-MILP, a hierarchical preference optimization framework for predict-and-search MILP solving. It introduces a holistic quality score that combines the objective value with the distance to a near-optimal convex hull of solutions, and constructs a three-tier preference hierarchy (near-optimal hull, high-quality feasible, perturbed). Training uses SimPO-style preference learning alongside BCE. Experiments on CA/SC/IP/WA and a MIPLIB subset show lower gaps and faster convergence than PS/ConPS and solvers.

**Strengths:**

- Clear motivation: objective-only labels are noisy; adding geometry via the convex hull distance is sensible.
- Method is well-instantiated: holistic score + three-tier preferences + SimPO integration.
- Strong empirical results with useful ablations (tier removal, pair count, temperature/margin, search params).

**Weaknesses:**

- Training/data prep overhead is not quantified: computing many near-optimal solutions, convex hull projection (QP via OSQP), and generating perturbed solutions can be expensive; it’s unclear whether the added cost is justified relative to the observed runtime gains.
- Preference pair acquisition on hard instances is under-explored: for very large problems or tiny feasible regions, obtaining diverse near-optimal sets and meaningful negative pairs may be difficult.
- Baselines are dated on the prediction side: newer predict-and-search variants and solution-prediction method.

**Questions:**

- Please report the end-to-end overhead: (i) number of solver calls per instance to collect near-optimal solutions; (ii) time for convex hull distance (QP) vs. Hamming fallback; (iii) total data-prep/training time vs. PS/ConPS. Are the test-time speed/quality gains worth the extra offline cost?

- How robust is preference-pair construction on harder instances with small feasible regions and >100k decision variables? Can you test on at least one large-scale case to assess scalability of hull distance, pair mining, and overall training stability?

- Will you add stronger, recent baselines and report under the same search hyperparameters?

---

### Official Review · Reviewer_whbN · 2025-11-01

**Soundness:** 3
**Presentation:** 2
**Contribution:** 2
**Rating:** 4
**Confidence:** 4

**Summary:**

This paper proposes HiPO-MILP, which introduces *Preference Optimization* into the process of using GNNs to predict initial solutions for MILP problems. The method employs three-tiered optimization signals as the optimization objectives in preference optimization.

**Strengths:**

1. The paper introduces Preference Optimization signals to guide the GNN training process for MILP optimization.
2. Experimental results show that HiPO-MILP outperforms the compared baseline methods.

**Weaknesses:**

1. HiPO requires the preparation of preference pairs prior to training, which introduces additional computational overhead compared with existing paradigms.
2. The paper should compare HiPO-MILP with more recent *predict-and-search* approaches for MILP solving, such as DiffILO.
3. While the paper introduces the concept of three-tier solutions through Hierarchical Preference Optimization (lines 270–281), the loss computation in Section 4.4 appears only weakly related to this three-tier structure.

**Questions:**

1. What is the computational cost of preparing the preference pairs? How is the solution list used to construct preference pairs obtained for each problem instance before training?
2. The paper acknowledges that the pair interval influences HiPO-MILP’s performance. How can one ensure that every MILP instance used in HiPO-MILP training has sufficiently many and high-quality preference pairs (e.g., a well-balanced distribution between preferred and rejected solutions)? If the feasible region of a problem is small, wouldn’t obtaining preference pairs become difficult?
3. Equation (6) requires computing $d(x,\text{conv}(S^*))$. Since this involves the convex hull of all optimal solutions $\text{conv}(S^*)$, does this imply that all optimal solutions of a MILP instance must be known when preparing preference pairs? This seems impractical—how is it actually implemented in HiPO-MILP?
4. Compared with existing work such as Contrastive Predict-and-Search (ConPS), what are the concrete algorithmic differences in HiPO-MILP beyond the introduction of the Holistic Quality Score?

---

### Note · Authors · 2025-11-24

I have read and agree with the venue's withdrawal policy on behalf of myself and my co-authors.